# Cytotoxic Potential of *Alternaria tenuissima* AUMC14342 Mycoendophyte Extract: A Study Combined with LC-MS/MS Metabolic Profiling and Molecular Docking Simulation

Amal A. Al Mousa [1,*][iD], Mohamed E. Abouelela [2,3,*][iD], Abdallah M. A. Hassane [4][iD], Fatimah S. Al-Khattaf [1], Ashraf A. Hatamleh [1][iD], Hadeel S. Alabdulhadi [5], Noura D. Dahmash [1] and Nageh F. Abo-Dahab [4][iD]

1   Department of Botany and Microbiology, College of Science, King Saud University, P.O. Box 145111, Riyadh 4545, Saudi Arabia
2   Department of Pharmacognosy, Faculty of Pharmacy, Al-Azhar University, P.O. Box 71524, Assiut 11651, Egypt
3   Department of Pharmaceutical Sciences, College of Pharmacy, University of Kentucky, 789 S. Limestone Street, Lexington, KY 40506, USA
4   Botany and Microbiology Department, Faculty of Science, Al-Azhar University, P.O. Box 71524, Assiut 11651, Egypt
5   Research Assistant Internship Program, Vice Rectorate for Graduate Studies and Scientific Research, King Saud University, Deanship of Scientific Research, Riyadh 4545, Saudi Arabia
*   Correspondence: aalmosa@ksu.edu.sa (A.A.A.M.); m_abouelela@azhar.edu.eg (M.E.A.); Tel.: +966-505389419 (A.A.A.M.); +20-1123565130 (M.E.A.)

**Abstract:** Breast, cervical, and ovarian cancers are among the most serious cancers and the main causes of mortality in females worldwide, necessitating urgent efforts to find newer sources of safe anticancer drugs. The present study aimed to evaluate the anticancer potency of mycoendophytic *Alternaria tenuissima* AUMC14342 ethyl acetate extract on HeLa (cervical cancer), SKOV-3 (ovarian cancer), and MCF-7 (breast adenocarcinoma) cell lines. The extract showed potent effect on MCF-7 cells with an IC50 value of 55.53 µg/mL. Cell cycle distribution analysis of treated MCF-7 cells revealed a cell cycle arrest at the S phase with a significant increase in the cell population (25.53%). When compared to control cells, no significant signs of necrotic or apoptotic cell death were observed. LC-MS/MS analysis of *Alternaria tenuissima* extract afforded the identification of 20 secondary metabolites, including 7-dehydrobrefeldin A, which exhibited the highest interaction score (−8.0156 kcal/mol) in molecular docking analysis against human aromatase. Regarding ADME pharmacokinetics and drug-likeness properties, 7-dehydrobrefeldin A, 4′-epialtenuene, and atransfusarin had good GIT absorption and water solubility without any violation of drug-likeness rules. These findings support the anticancer activity of bioactive metabolites derived from endophytic fungi and provide drug scaffolds and substitute sources for the future development of safe chemotherapy.

**Keywords:** cancer; *Alternaria tenuissima*; cytotoxicity; LC-MS/MS; aromatase; molecular docking

## 1. Introduction

Human health is at constant risk due to the occurrence of different types of non-communicable chronic diseases [1]. Among the major non-communicable chronic diseases, cancer is considered the second principal cause of mortality in the world [2]. In 2020, an estimated 19.3 million cancer cases developed and around 10.0 million cancer deaths occurred worldwide [3]. Breast cancer was diagnosed in 2.3 million new cases (11.7%), followed by 11.4% for lung, 10.0% for colorectal, 7.3% for prostate, and 5.6% for stomach cancers [3]. Cancer incidence could be initiated by both extrinsic and intrinsic factors that trigger the activation or inactivation of certain genes, subsequently leading to abnormal growth of cells [4].

The massive replicative potential and loss of balance between cell proliferation and apoptosis are attributes that increase the failure of damaged cells to be eliminated via apoptosis [5,6]. Many natural products considered to be potential sources of novel anticancer drugs exert antitumor effects by inhibiting cell proliferation and apoptosis [7]. Most of the anticancer therapeutic agents have unpleasant side effects. As a result, researchers are attempting to develop new anticancer drugs that are more targeted to abnormally proliferating cancerous cells while having minimal effects on normal cells [1,8]. However, the development of new drugs from natural sources with fewer side effects is emerging as a promising area of research in cancer [9].

It is well known that estrogens are required for the growth of breast cancer in both premenopausal and postmenopausal women who have an elevated risk of occurrence [10]. Thus, depression of estrogen levels by inhibiting aromatase activity is a promising effective therapeutic target for breast cancer treatment [11,12]. Aromatase is a crucial enzyme that converts androgen precursors into estrogen. Aromatase inhibitors are an effective targeted therapy in patients with estrogen-dependent breast cancer. They are one of two popular remediation tactics for breast cancer treatment with estrogen receptor modulation by selective estrogen receptor modulators [13]. A total of 282 natural compounds of diverse categories including flavonoids, alkaloids, terpenoids, lignans, xanthones, peptides, and fatty acids have been evaluated for their inhibitory potential against aromatase [14].

Recently, natural products from endophytic fungi have been identified as an untapped valuable source of many new biologically active chemical analogs against infectious diseases, cancer, and a variety of other diseases [15–17]. Endophytic fungi revealed the presence of different classes of compounds, including alkaloids, steroids, phenols, quinones, coumarins, flavonoids, and xanthones [17,18]. Early reports proposed that naturally bioeffective mycoendophytic metabolites could be utilized as a novel source of anticancer chemotherapeutic discovery [19]. Mycoendophytes have been found to fabricate a diverse range of biopotent active secondary products, including paclitaxel, torreyanic acid, podophyllotoxin, camptothecin, and vincristine, which are well known for their antiproliferative and antitumor properties [1,20].

*Alternaria tenuissima* is a species of the *Alternaria* genus that belongs to the Dematiaceae family of fungi that inhabit various plant species and have a wide distribution in the environment [21,22]. *Alternaria* species revealed several categories of secondary metabolites which include nitrogen-containing compounds, polyketides, steroids, terpenoids, pyranones (pyrones), quinones, and phenolics [23,24]. In addition, it has a number of biological activities, including phytotoxic, cytotoxic, and antimicrobial properties [25,26].

Computational chemistry methods have been successfully applied to investigate the chemical reactions and binding patterns of a broad variety of biological and chemical systems [27,28]. Moreover, the calculation of the binding capacity of a compound consumes time and cost in in vivo and in vitro drug studies [29]. As a result, in silico tools such as molecular docking are effectively used for providing insight into the binding as well as interaction strengths of ligand inhibitors for drug discovery, which involves a target-based approach based on the target and ligand structure, thus saving time, effort, and cost as well [30–33].

In the current study, we evaluate the effect of an ethyl acetate extract from an *Alternaria tenuissima* AUMC14342 endophytic strain on HeLa (cervical cancer), SKOV-3 (ovarian cancer), and MCF-7 (breast adenocarcinoma), as well as the extract's bioactive constituent profile using LC-MS/MS metabolomic techniques. In addition, we investigated the molecular simulation of the tentatively identified compounds against human placental aromatase cytochrome P450 along with the ADME pharmacokinetics of the highest affinity compounds.

## 2. Materials and Methods

### 2.1. Culturing, Propagation, and Extraction of Alternaria tenuissima

Mycelial discs (5 mm diameter) from a 7-day-old potato dextrose agar culture of endophytic fungus *Alternaria tenuissima* AUMC14342 isolated from *Artemisia judaica* L. [34]

culture were inoculated into sterilized rice medium in 1 L Erlenmeyer flasks and incubated for 30 days at 28 ± 2 °C. The fermented culture was extracted twice by using ethyl acetate (EtOAc) [34], filtrated, and evaporated by a rotary evaporator at 45 °C to produce the dry extract that was kept for further investigations.

## 2.2. Cell Culture and Cytotoxicity Assay

The cancer cell lines were obtained from Nawah Scientific Inc. (Mokatam, Cairo, Egypt). HeLa (cervical cancer) and SKOV-3 (ovarian cancer) cells were cultured in RPMI medium amended with 100 units/mL penicillin, 100 mg/mL streptomycin, and 10% heat-inactivated fetal bovine serum and incubated in a humidified 5% $CO_2$ atmosphere at 37 °C, while MCF-7 (breast adenocarcinoma) cells were grown in DMEM medium [31]. The Sulforhodamine B (SRB) assay was employed to estimate cancer cell viability, and a BMG LABTECH FLUOstar Omega microplate reader (Ortenberg, Germany) was utilized to measure absorbance at 540 nm [35,36].

## 2.3. Effect of A. tenuissima Extract on Cell Cycle Distribution of Breast Cancer Cells

After treatment with test compounds for 24 h, cells were analyzed for DNA content using a flow cytometry analysis protocol [37] with the FL2 (λex/em 535/617 nm) signal detector (ACEA Novocyte flow cytometer, ACEA Biosciences Inc., San Diego, CA, USA). Twelve thousand events were obtained for each tested sample. ACEA NovoExpress software (ACEA Biosciences Inc., San Diego, CA, USA) was utilized to calculate cell cycle distribution [38].

## 2.4. Assessment of Apoptotic Effect of A. tenuissima Extract

An Annexin-V/FITC kit (Abcam Inc., Cambridge Science Park, Cambridge, UK) for apoptosis detection combined with 2-fluorescent-channel flow cytometry was used to determine apoptotic and necrotic cell populations as previously described [35,37].

## 2.5. UHPLC–ESI–MS/MS Profiling

The UHPLC–ESI–MS/MS analysis of the ethyl acetate extract of *A. tenuissima* was performed on an ExionLC AC system coupled with a SCIEX Triple Quad 5500 + MS/MS system equipped with an electrospray ionization (ESI) system. An Ascentis C18 Column (4.6 × 150 mm, 3 μm) was employed as the stationary phase, and the sample was eluted with mobile phases consisting of eluent A (0.1% formic acid) and eluent B (acetonitrile, LC grade) with following mobile phase gradient: 10% B at 0–1 min, 10%–90% B at 1–21 min, 90% B at 21–25 min, and 10% at 25.01–28 min. The flow rate was 0.5 mL/min, and the injection volume was 10 μL. MS/MS analysis used positive and negative ionization modes with a scan (EMS-IDA-EPI) [35]. The compounds were identified by using MS-DIAL version 4.70, Natural Products Atlas, and CFM-ID version 4.0 software [39–42].

## 2.6. Molecular Docking Simulation

The binding affinity of the identified metabolites was evaluated by molecular docking analysis in comparison with a standard reference inhibitor using the "Molecular Operating Environment (MOE 2014.09) [35]. The compounds were imported to MOE and subjected to 3D protonation and Merck molecular force field (MMFF94x) energy minimization, and they were partially charged. A stochastic conformational search was conducted, the minimum dE conformers were selected, and a virtual ligand database was constructed [43]. The structure of human placental aromatase cytochrome P450 (CYP19A1) (PDB ID: 3S79) was acquired from the Protein Data Bank [33]. All the hetero atoms and unbound water molecules were removed from the target proteins, and their structures were optimized for docking simulation. The parameters of scoring were Triangle Matcher, scoring was set at London dG and rescoring at GBVI/WSA dG, and the docking poses were set at 30 poses for the initial energy score and 10 for refinement [43]. The docking pose score (the process of evaluating a particular pose by counting the number of favorable intermolecular interactions such as

hydrogen bonds and hydrophobic contacts and computed by summing all the applicable scores of any interacting surface points between cavity and ligand) [44], root mean square deviation (RMSD) (RMSD measures the difference in conformation and position between two poses of a molecule), and ligand–receptor complexes were tested for interaction analysis. The 3D images were created using the MOE visualizing tool [33,35]. The protocol was validated after protein preparation by running redocking of the complexed inhibitor to the active site, and the RSMD value was 0.18 Å. Complexed ligand and redocked ligand overlays are shown in the Supplementary Materials (Supplementary Figure S1).

### 2.7. Drug-like Properties and ADME Prediction of High-Affinity Compounds

The Molinspiration web server was used to determine the molecular properties [45,46], and the SwissADME website server was employed to calculate the drug-likeness, ADME, and pharmacokinetic parameters of the identified metabolites [33,42].

## 3. Results

### 3.1. Cytotoxic Activity of A. tenuissima Ethyl Acetate Extract

In the current study, the ethyl acetate extract of *A. tenuissima* was tested for its effect on the cell viability of three cancer cell lines, namely HeLa, SKOV-3, and MCF-7, by SRB assay, and the $IC_{50}$ values were determined from dose–response curves of different concentrations (Figure 1).

The results showed that the extract exhibited potential cytotoxic activity against the tested cell lines and had a prominent influence against the MCF-7 cell line, with the $IC_{50}$ value of 55.53 μg/mL (Table 1), in comparison with doxorubicin as a standard anticancer drug.

**Table 1.** $IC_{50}$ values of *A. tenuissima* extract and doxorubicin for tested cancer cell lines.

| Treatment | $IC_{50}$ (μg/mL) | | |
|---|---|---|---|
| | HeLa | SKOV-3 | MCF-7 |
| *A. tenuissima* **EtOAc extract** | $67.76 \pm 1.54$ | $74.60 \pm 1.55$ | $55.53 \pm 1.22$ |
| **Doxorubicin** | $0.05 \pm 0.07$ | $0.04 \pm 0.09$ | $0.34 \pm 0.43$ |

The histopathology study of the extract at a concentration of 100 μg/mL showed consistency with the dose–response curve (Figures 1 and 2) and $IC_{50}$ results (Table 1). Figure 2 shows the effect of the extract on different cell lines in comparison with the control and doxorubicin treatment. The observation of the optical microscope staining image changes of cancer cells showed that the control cells had normal morphology and attachment, while the cells treated with *A. tenuissima* EtOAc extract showed a significant reduction in cells at 100 μg/mL, confirming the cytotoxicity of the tested extract.

### 3.2. Effect of A. Tenuissima Extract on Cell Cycle Distribution of MCF-7 Cells

Breast cancer cells (MCF-7) were manipulated for 24 h with the predetermined $IC_{50}$ (55.53 μg/mL) of the *A. tenuissima* EtOAc extract, and the DNA content was measured using flow cytometry to determine the effect of the tested sample on MCF-7 cell cycle distribution (Figure 3).

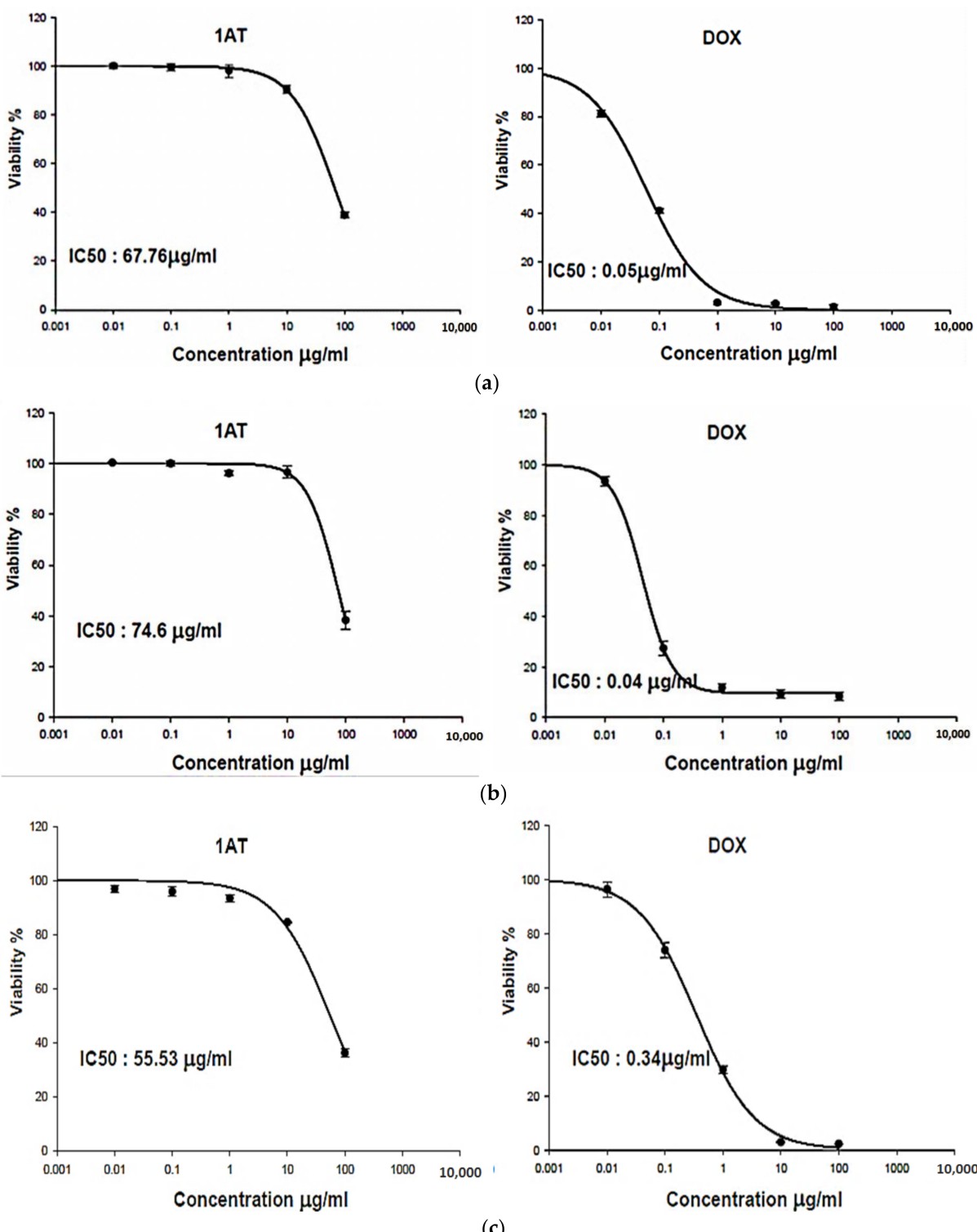

**Figure 1.** IC$_{50}$ dose–response curves of cytotoxic activities of *A. tenuissima* extract and doxorubicin against (**a**) HeLa cell line, (**b**) SKOV-3 cell line, (**c**) MCF-7 cell line.

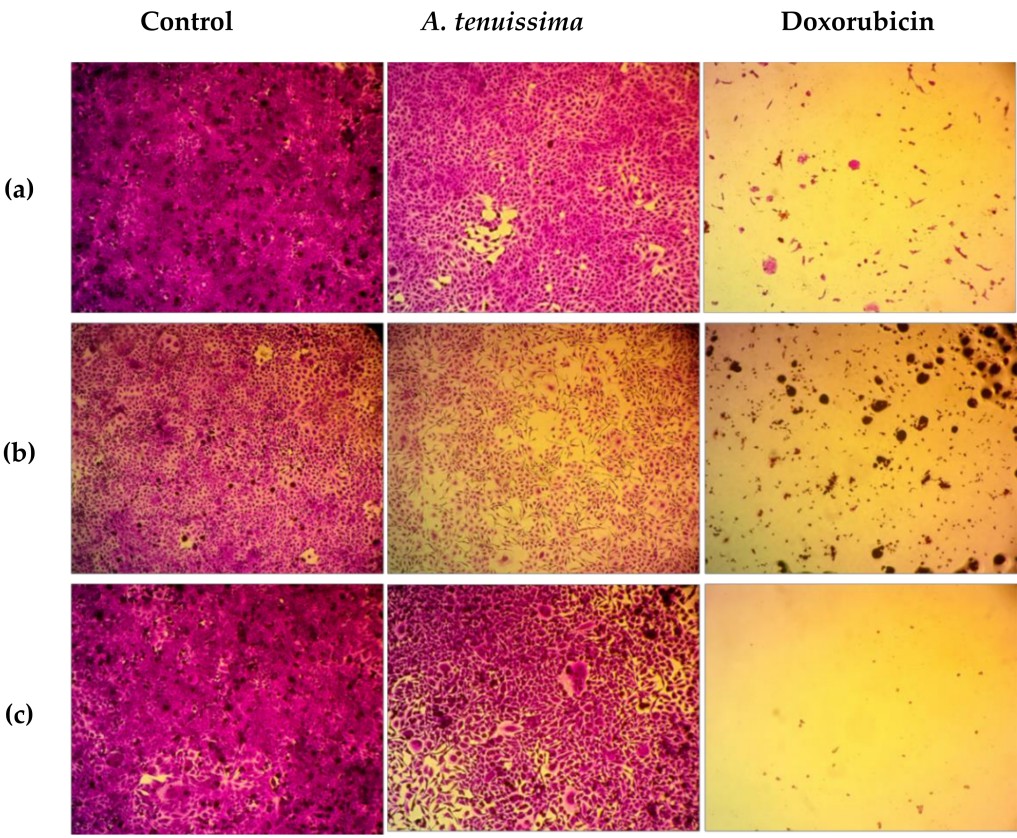

**Figure 2.** Optical microscope images of cytotoxicity assay *A. tenuissima* extract and doxorubicin (100 µg/mL) on HeLa cells (**a**), SKOV-3 (**b**), and MCF-7 (**c**), magnification power: X100.

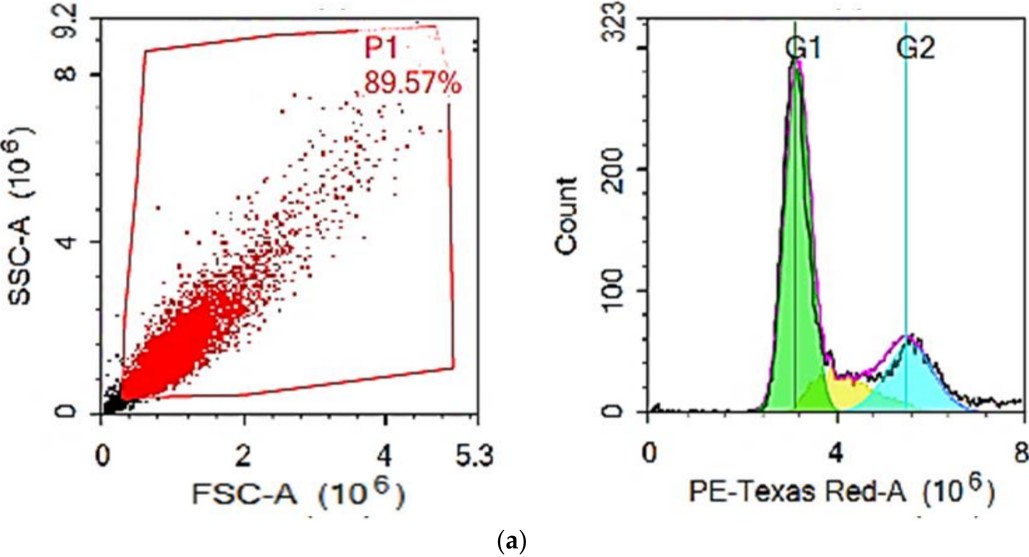

(**a**)

**Figure 3.** *Cont.*

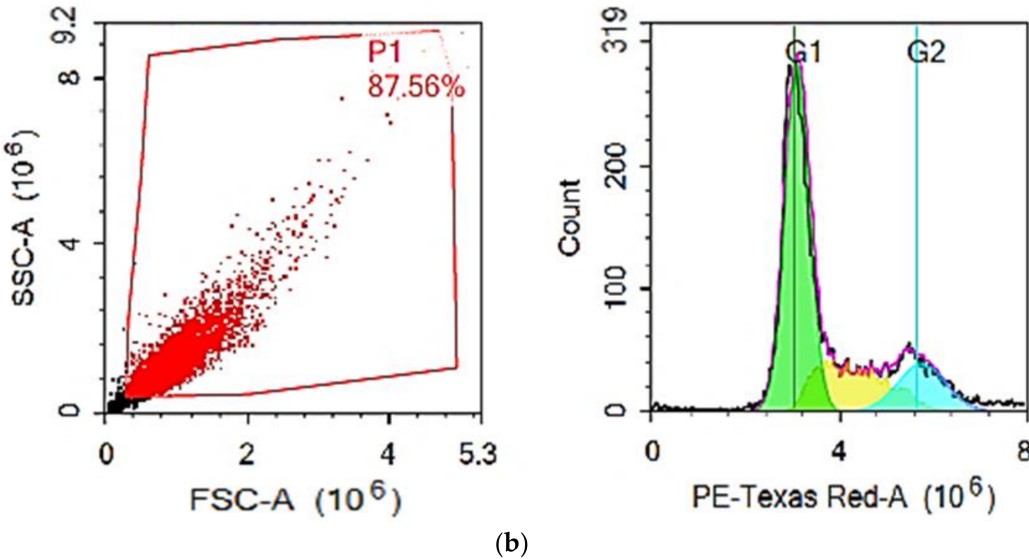

(**b**)

**Figure 3.** Cell cycle distribution of MCF-7 cells: (**a**) control; (**b**) *A. tenuissima* EtOAc extract.

With no discernible effects on other phases, the tested extract caused S-phase arrest and increased this cell population significantly, by 25.53%, from 16.80% to 22.56% (Table 2, Figures 3 and 4), suggesting that the effect on the DNA synthesis step of replication may be the cause of the extract's cytotoxic effect.

**Table 2.** Cell cycle distribution of MCF-7 cells.

| Treatment | Distribution (%) | | | |
|---|---|---|---|---|
| | **G1 Phase** | **S Phase** | **G2/M** | **Sub-G1** |
| **Control** | $58.80 \pm 1.95$ | $16.80 \pm 1.64$ | $22.37 \pm 0.06$ | $0.98 \pm 0.49$ |
| ***A. tenuissima* EtOAc extract** | $58.93 \pm 0.78$ | $22.56 \pm 0.18$ | $16.25 \pm 0.58$ | $1.48 \pm 0.35$ |

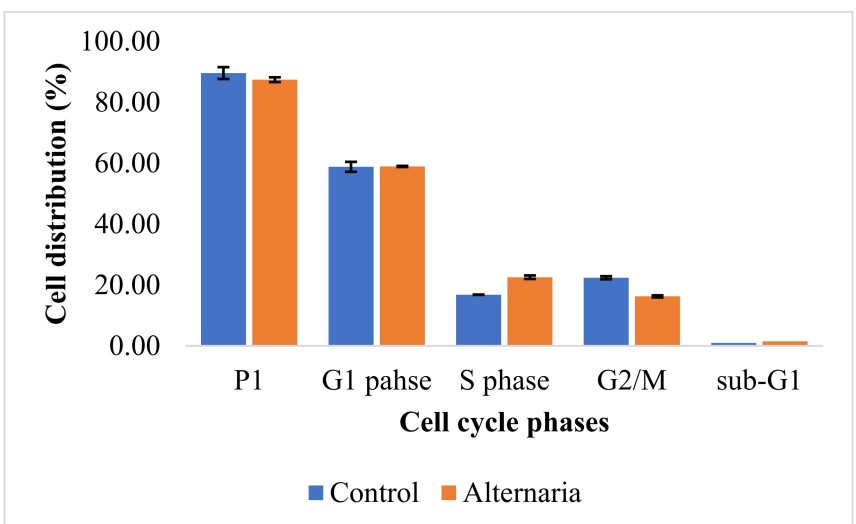

**Figure 4.** Comparison between cell cycle distribution phases of MCF-7 cells: control; *A. tenuissima* EtOAc extract.

*3.3. Assessment of Apoptotic Effect of A. tenuissima Extract*

After being exposed to the predetermined $IC_{50}$'s, cells were evaluated using Annexin-V/FITC staining in conjunction with flow cytometry to determine the impact of the extract on the mechanism of cell death. After 24 h, the evaluated extract showed no significant

signs of necrotic or apoptotic cell death when compared to untreated control cells (Table 3, Figure 5).

**Table 3.** Apoptosis/necrosis assessment in MCF7 cell line after exposure to *A. tenuissima* EtOAc extract and control for 24 h.

|  | Control (%) | *A. tenuissima* EtOAc Extract (%) |
| --- | --- | --- |
| **Q2-1** | $0.34 \pm 0.05$ | $0.39 \pm 0.02$ |
| **Q2-2** | $0.13 \pm 0.04$ | $0.04 \pm 0.05$ |
| **Q2-3** | $99.44 \pm 0.06$ | $99.56 \pm 0.05$ |
| **Q2-4** | $0.10 \pm 0.03$ | $0.00 \pm 0.00$ |

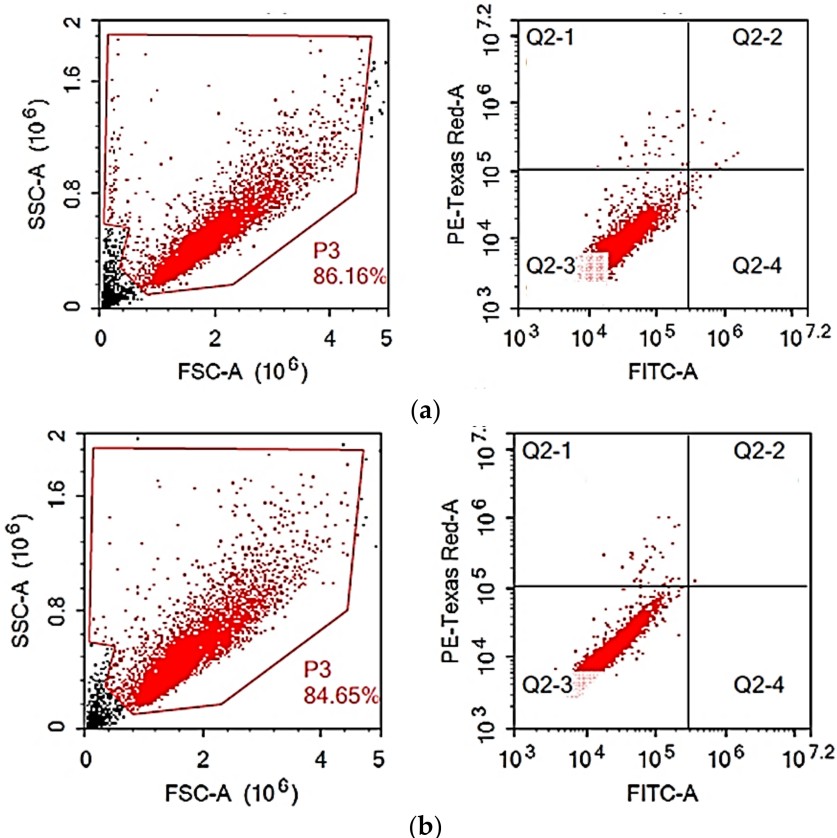

**Figure 5.** Apoptosis/necrosis estimation in MCF7 cells after treatment with *A. tenuissima* extract for 24 h: (**a**) control; (**b**) *A. tenuissima* EtOAc extract.

### 3.4. Metabolite Analysis of A. tenuissima EtOAc by UPLC-ESI-MS/MS

The analysis of the sample was performed using LC-ESI-MS/MS for the separation and detection of metabolites. Positive and negative ionization modes were employed to characterize the corresponding signals. The total ion current map of the sample was produced. The TIC of the *A. tenuissima* EtAOc extract is shown in Figure 6.

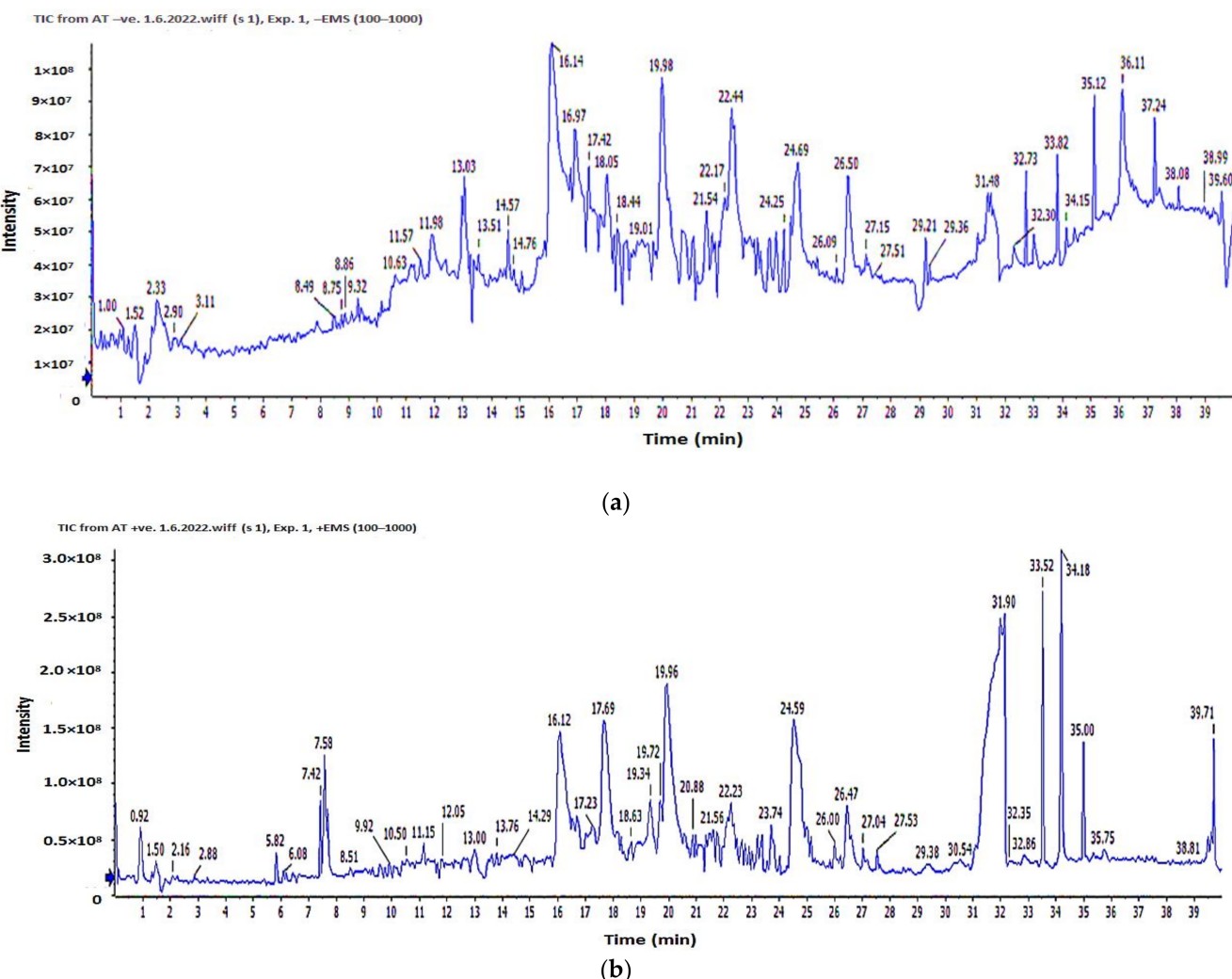

**Figure 6.** ESI-TIC chromatograms of *A. tenuissima* ethyl acetate extract: (**a**) negative mode; (**b**) positive mode.

Structural analysis was performed and the components were analyzed using two-stage mass spectrometry to obtain the mass, metabolite molecular formula, and characteristic fragment ions. Moreover, the previously isolated compounds of the *Alternaria* genus were used as a tool for the identification of detected compounds by comparison of the obtained molecular formula with the published data.

In total, 20 secondary metabolites were identified by analysis of the results (Table 4, Figure 7) based on their precursor ion and their MS$^2$ fragmentation patterns compared with the Competitive Fragmentation Modeling for Metabolite Identification online database; the retention time, high-resolution molecular ion mass, chemical formula, and MS$^2$ fragments of the identified chemical constituents are shown in Table 4 and Supplementary Figures S2–S21. The compounds are arranged according to their retention time.

**Table 4.** List of tentatively identified compounds in *A. tenuissima* extract.

| No. | Name | $t_R$ | MF | [M−H]⁻/[M+H]⁺ | MS² Fragments |
|---|---|---|---|---|---|
| 1 | Cyclo-Ala-Pro-Diketopiperazine | 4.49 | $C_8H_{12}N_2O_2$ | 167/- | 165, 151,149, 139, 68 |
| 2 | (S)-alternariphent A | 12.43 | $C_{13}H_{14}O_4$ | 233/- | 217, 215, 205, 189, 161, 147, 123 |
| 3 | Alternapyran | 12.84 | $C_{10}H_{10}O_2$ | 161/- | 143, 133, 117 |
| 4 | Altenuene (4′-Epialtenuene) | 14.32 | $C_{15}H_{16}O_6$ | 291/- | 273, 261, 259, 247, 245, 205, 203 |
| 5 | Solanapyrone G | 14.71 | $C_{17}H_{21}NO_3$ | 286/- | 284, 268, 258, 173 |
| 6 | 4-methoxy-6-methyl-5-(3-oxobutyl)-2H-pyran-2-one | 15.04 | $C_{11}H_{14}O_4$ | 209/- | 191, 177, 181, 139, 109 |
| 7 | N-acetyltyramine | 15.83 | $C_{10}H_{13}NO_2$ | -/180 | 162, 136, 121, 107 |
| 8 | Versimide | 16.04 | $C_9H_{11}NO_4$ | 196/- | 168, 142, 138, 136, 112 |
| 9 | 7-dehydrobrefeldin A | 16.21 | $C_{16}H_{22}O_4$ | -/279 | 261, 243, 237, 217, 201 |
| 10 | Alternariol | 16.84 | $C_{14}H_{10}O_5$ | 257/- | 215, 189, 173 |
| 11 | 5-butyl-4-methoxy-6-methyl-2H-pyran-2-one | 17.21 | $C_{11}H_{16}O_3$ | -/197 | 153, 139, 111, 97 |
| 12 | Altertenuol | 18.03 | $C_{14}H_{10}O_6$ | -/275 | 257, 229, 219, 177, 201 |
| 13 | 3-O-demethylaltenuisol | 18.08 | $C_{13}H_8O_6$ | 259/- | 241, 231, 215, 187, 175, 161, 135 |
| 14 | (-)-alternarlactam | 18.85 | $C_{14}H_{13}NO_4$ | 258/- | 240, 230, 214, 210, 174, 108 |
| 15 | 2-(N-vinylacetamide)-4-hydroxymethyl-3-ene-butyrolactone | 19.77 | $C_8H_9NO_4$ | -/184 | 166, 154, 142, 125, 122 |
| 16 | Alternariol-9-methyl ether | 25.34 | $C_{15}H_{12}O_5$ | 271/- | 256, 229 |
| 17 | Resveratrodehyde C | 30.02 | $C_{15}H_{12}O_4$ | 255/- | 237, 201, 197, 185, 135 |
| 18 | Solanapyrone P | 30.23 | $C_{16}H_{22}O_3$ | -/263 | 245, 219, 205, 161 |
| 19 | Atransfusarin | 31.13 | $C_{13}H_{18}N_2O_4$ | 265/- | 219, 178, 175, 161, 150, 106 |
| 20 | Alternatain A | 32.17 | $C_{14}H_{16}O_6$ | -/281 | 263, 245, 207, 179, 171, 151 |

Analyzing the *A. tenuissima* extract revealed several compounds (Table 4) in positive ion mode. The molecular ion mass peaks at *m/z* 180 and 263 [M+H]+ for the predicted molecular formulas $C_{10}H_{13}NO_2$ and $C_{16}H_{22}O_3$ gave hits of N-acetyltyramine and solanapyrone P, respectively. The mass ion peaks at *m/z* 279, 197, 275, 184, and 281 were fit with 7-dehydrobrefeldin A, 5-butyl-4-methoxy-6-methyl-2H-pyran-2-one, altertenuol, 2-(N-vinylacetamide)-4-hydroxymethyl-3-ene-butyrolactone, and alternatain A, respectively.

The negative mode results (Table 4) revealed the presence of nitrogen-containing compounds at $t_R$ 4.49, 14.71, 16.04, 18.85, and 31.13 min corresponding to molecular formulas $C_8H_{12}N_2O_2$, $C_{17}H_{21}NO_3$, $C_9H_{11}NO_4$, $C_{14}H_{13}NO_4$, and $C_{13}H_{18}N_2O_4$ assigned for cyclo-ala-pro-diketopiperazine, solanapyrone G, versimide, (-)-alternarlactam, and atransfusarin, respectively. Moreover, the mass ion peaks at *m/z* 291, 257, 259, and 271 corresponding to the suggested molecular formulas $C_{15}H_{16}O_6$, $C_{14}H_{10}O_5$, $C_{13}H_8O_6$, and $C_{15}H_{12}O_5$ [M−H]⁻ fit to isocoumarin derivatives altenuene (4′-epialtenuene), alternariol, 3-O-demethylaltenuisol, and alternariol-9-methyl ether. In addition, (S)-alternariphent A, alternapyran, 4-methoxy-6-methyl-5-(3-oxobutyl)-2H-pyran-2-one, and resveratrodehyde C were also detected at $t_R$ 12.43, 12.84, 15.04, and 30.02, respectively.

**Figure 7.** Chemical structures of detected compounds in *A. tenuissima* EtOAc extract.

### 3.5. Molecular Docking Study

The compounds' docking results revealed an affinity range of −8.0156 to −4.7618 kcal/mol. The top-scoring compounds were 4'-epialtenuene (4'), 7-dehydrobrefeldin A (9), and atrans-fusarin (19), which had a stronger affinity to the human placental aromatase cytochrome P450 (CYP19A1) active site relative to the complexed inhibitor ligand 4-androstene-3-17-dione (Table 5).

**Table 5.** Docking score of identified compounds against human placental aromatase cytochrome P450 (CYP19A1) (PDB ID: 3S79).

| No. | Compound | Pose Score (kcal/mol) | RMSD * Refine (Å) | E. Conf.** (kcal/mol) |
|---|---|---|---|---|
| 1 | Cyclo-Ala-Pro-Diketopiperazine | −4.7618 | 1.00 | 51.04 |
| 2 | (S)-alternariphent A | −6.0536 | 0.90 | 43.64 |
| 3 | Alternapyran | −4.7764 | 1.09 | 27.17 |
| 4 | Altenuene | −6.3639 | 1.64 | 59.34 |
| 5 | 4′-Epialtenuene | −7.2929 | 1.30 | 59.64 |
| 6 | Solanapyrone G | −6.7726 | 1.04 | 38.47 |
| 7 | 4-methoxy-6-methyl-5-(3-oxobutyl)-2H-pyran-2-one | −6.3077 | 1.01 | 6.33 |
| 8 | Nacetyltyramine | −5.8608 | 0.74 | −24.54 |
| 9 | Versimide | −5.0443 | 1.48 | −16.74 |
| 10 | 7-dehydrobrefeldin A | −8.0156 | 1.16 | 9.92 |
| 11 | Alternariol | −6.7608 | 1.13 | 36.32 |
| 12 | 5-butyl-4-methoxy-6-methyl-2H-pyran-2-one | −6.1478 | 0.99 | 13.38 |
| 13 | Altertenuol | −6.1731 | 0.70 | 46.44 |
| 14 | 3-O-demethylaltenuisol | −6.3409 | 0.88 | 21.13 |
| 15 | (-)-alternarlactam | −6.3888 | 1.84 | 20.48 |
| 16 | 2-(N-vinylacetamide)-4-hydroxymethyl-3-ene-butyrolactone | −5.2035 | 1.66 | 3.83 |
| 17 | Alternariol-9-methyl ether | −6.6327 | 1.70 | 55.56 |
| 18 | Resveratrodehyde C | −6.5303 | 1.39 | 11.96 |
| 19 | Solanapyrone P | −6.6332 | 0.78 | 20.16 |
| 20 | Atransfusarin | −7.2688 | 1.49 | 46.40 |
| 21 | Alternatain A | −6.0734 | 1.28 | 77.03 |
| 22 | 4-androstene-3-17-dione | −9.0034 | 1.09 | 55.94 |

* RMSD, root mean square deviation. ** E. Conf., energy of conformer.

Molecular docking analysis revealed that 7-dehydrobrefeldin A (9) showed the highest interaction score with human aromatase (posing score -8.0156 kcal/mol and RMSD value of 1.16 Å) as compared to 4-androstene-3-17-dione, standard inhibitor (−9.0037 kcal/mol). 7-dehydrobrefeldin A interacts through hydrogen bond formation with the Met 374 amino acid residue in the catalytic cleft of aromatase in addition to several hydrophobic interactions with the active site (Figure 8).

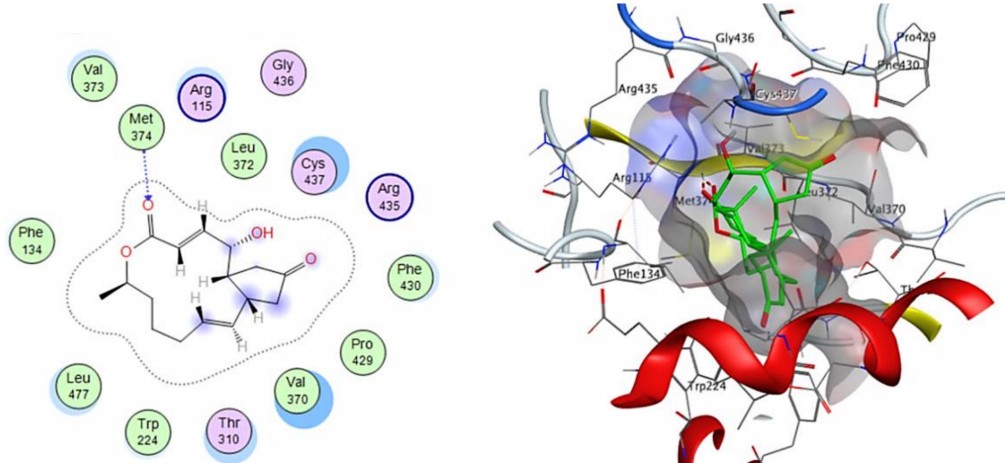

**Figure 8.** The 2D and 3D interactions of 7-dehydrobrefeldin A with human aromatase cytochrome P450 active site.

Moreover, the compound 4′-Epialtenuene showed a pose score of −7.2929 kcal/mol with an RMSD value of 1.30 Å and was found to bind with the receptor active site through the formation of hydrogen bonds with Leu 477 and Met 374 amino acid residues as H-donors as well as through a pi-H bond with VAL 370 amino acid (Figure 9).

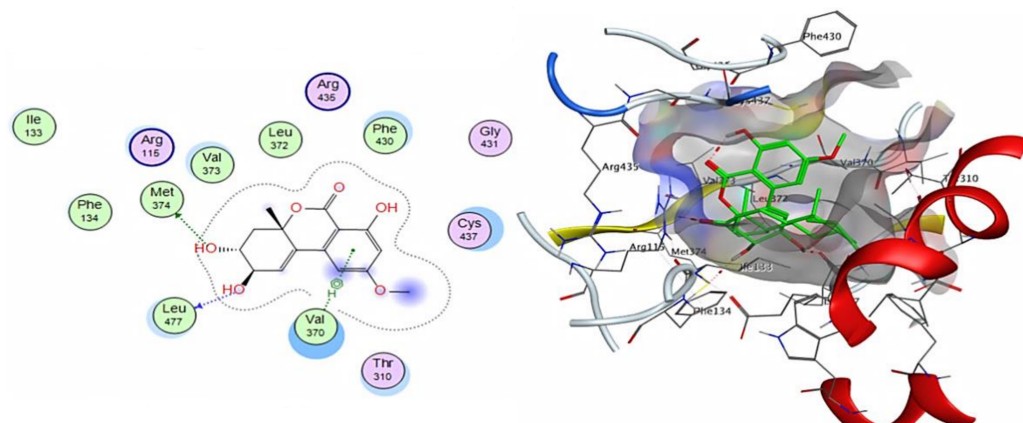

**Figure 9.** The 2D and 3D interactions of 4′-epialtenuene with human aromatase cytochrome P450 active site.

Furthermore, atransfusarin (19) interacts with Met 374 amino acid residue as an H-donor and with VAL 370 amino acid through a pi-H bond (Figure 10) with a pose score of −7.2688 kcal/mol. It could be concluded that these compounds could be used as a scaffold for the development of bioactive treatment for breast cancer through inhibition of the selected target (human aromatase cytochrome P450)

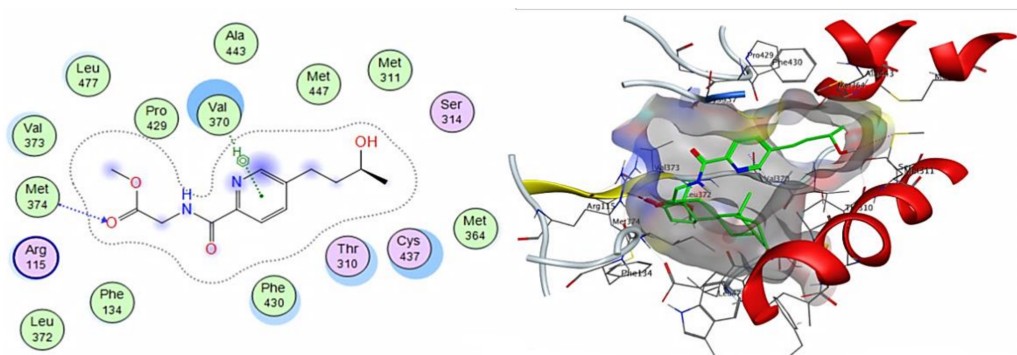

**Figure 10.** The 2D and 3D interactions of atransfusarin with human aromatase cytochrome P450 active site.

### 3.6. ADME Pharmacokinetics and Drug-Likeness Properties of High-Affinity Compounds

The compounds that showed the highest affinity to the human aromatase cytochrome P450 active site were screened for their drug-likeness and ADME pharmacokinetics using the website servers. All tested compounds had good GIT absorption and water solubility without any violation of drug-likeness rules. In addition, the compounds had promising bioavailability scores with a value of 0.55. 4′-epialtenuene and dehydrobrefeldin A showed promising lead-likeness and synthetic accessibility (Table 6).

**Table 6.** Detailed in silico assessment of molecular properties, drug-likeness, absorption, distribution, metabolism, and excretion of 4′-epialtenuene, 7-dehydrobrefeldin A, and atransfusarin.

| Molecule | 4′-Epialtenuene | 7-Dehydrobrefeldin A | Atransfusarin |
|---|---|---|---|
| Molecular weight | 292.28 g/mol | 278.34 g/mol | 266.29 g/mol |
| TPSA | 96.22 | 63.6 | 88.52 |
| MLOGP | 0.6 | 1.78 | −0.05 |
| #Heavy atoms (natoms) | 21 | 20 | 19 |
| #Aromatic heavy atoms | 6 | 0 | 6 |
| nON | 6 | 4 | 6 |
| nOHNH | 3 | 1 | 2 |
| #Rotatable bonds | 1 | 0 | 8 |
| Fraction Csp3 | 0.4 | 0.62 | 0.46 |
| #H-bond acceptors | 6 | 4 | 5 |
| #H-bond donors | 3 | 1 | 2 |
| Molecular volume | 250.87 | 268.63 | 247.61 |
| Lipinski #violations | 0 | 0 | 0 |
| Ghose #violations | 0 | 0 | 0 |
| Veber #violations | 0 | 0 | 0 |
| Egan #violations | 0 | 0 | 0 |
| Muegge #violations | 0 | 0 | 0 |
| ESOL Class | Soluble | Soluble | Very soluble |
| GI absorption | High | High | High |
| BBB permeant | No | Yes | No |
| Pgp substrate | No | No | No |
| CYP1A2 inhibitor | No | No | No |
| CYP2C19 inhibitor | No | No | No |
| CYP2C9 inhibitor | No | No | No |
| CYP2D6 inhibitor | No | No | No |
| CYP3A4 inhibitor | No | No | No |
| Bioavailability Score | 0.55 | 0.55 | 0.55 |
| Lead-likeness #violations | 0 | 0 | 1 |
| Synthetic Accessibility | 4.26 | 4.87 | 2.65 |

## 4. Discussion

Fungi are a valuable source of bioefficient natural products that can be employed to fabricate new analogs for cancer treatment [5]. Several metabolites from *Alternaria* spp. of various chemical classes and biological activities have been identified [23]. *Alternaria* extracts and secondary metabolites have been shown to have a wide range of biological activities and functions, including cytotoxic, antimicrobial, and herbicide effects. The promising bioactivities have attracted the interest of pharmaceutical researchers in the discovery of novel natural therapeutics [23,34].

In the present study, the $IC_{50}$ values of the *A. tenuissima* EtOAc extract on the viability of HeLa, SKOV-3, and MCF-7 cancer cell lines determined by SRB assay were 67.76, 74.60, and 55.53 µg/mL, respectively. Cell cycle distribution analysis of the treated MCF-7 cell line showed a cell cycle arrest at the S phase with a significant increase in the cell population (25.53%), while no significant signs of necrotic or apoptotic cell death were observed. It is noteworthy that the previous studies of *A. tenuissima* revealed the isolation of toxic metabolites such as alternariol and alternariol monomethyl ether which possessed strong cytotoxicity against KB and KBv200 tumor cell lines [47] and L5178Y mouse lymphoma cells [48]. Interestingly, Lehmann et al. reported the estrogenic potential of alternariol, as well as its cell proliferation inhibitory effect via interference with the cell cycle [49]. Furthermore, tenuazonic acid, a metabolite isolated from several endophytic *Alternaria* species, exhibited a broad spectrum of biological activity, such as antineoplastic, antiviral, and antibiotic effects [24,50,51]. Additionally, *A. tenuissima* TER995 was employed as a promising source for the submerged fermentation-based production of paclitaxel (taxol), the most valuable anticancer drug [52].

Analyzing the *A. tenuissima* extract in positive ion mode revealed several compounds, including N-acetyltyramine and solanapyrone P, which were previously isolated from *A. tenuissima* SY-P-07 and SP-07 [53,54]. Moreover, 7-dehydrobrefeldin A from *A. carthami* [55], 5-butyl-4-methoxy-6-methyl-2H-pyran-2-one from A. phragmospora [56], altertenuol from *Alternaria* sp. [57], 2-(N-vinylacetamide)-4-hydroxymethyl-3-ene-butyrolactone from *Alternaria* sp. NH-F6 [58], and alternatain A from *A. alternata* MT-47 [59] were detected. The negative mode results revealed the presence of previously obtained cyclo-ala-pro-diketopiperazine from *A. tenuissima* [53], solanapyrone G from *A. solani* [60], versimide from *A. tenuis* [61], (-)-alternarlactam from *Alternaria* sp. HG1 [62], atransfusarin from *A. atrans* MP-7 [63], altenuene (4′-epialtenuene) from *Alternaria* sp. and *A. tenuis* [48,64], alternariol from *A. tenuis* [65], alternariol-9-methyl ether from *A. tenuissima* [65], 3-O-demethylaltenuisol from *Alternaria* sp. PfuH1 [66], (S)-alternariphent A from *Alternaria* sp. SCSIO41014 [67], alternapyran from *Alternaria* sp. ZG22 [68], 4-methoxy-6-methyl-5-(3-oxobutyl)-2H-pyran-2-one from *A. phragmospora* [56], and resveratrodehyde C from *Alternaria* sp. R6 [69].

The cytotoxicity assay revealed that *A. tenuissima* has a potent impact on MCF-7 cells, with an IC$_{50}$ value of 55.53 μg/mL. In order to evaluate the possible affinity of identified compounds and understand their binding possibility as inhibitors for the aromatase enzyme as a selective target for breast cancer treatment, a molecular docking analysis was conducted [13,33]. Aromatase is responsible for estrogen biosynthesis by catalyzing the bioconversion of androgens to estrogens by the aromatization of androstenedione to estrone [27,70]. Aromatase inhibitors are classified into steroidal and non-steroidal types depending on their chemical structure [71]. Both steroidal and non-steroidal inhibitors of aromatase block the biosynthesis of estrogens by inhibiting aromatase; steroidal inhibitors inhibit aromatase in an irreversible manner, while non-steroidal inhibitors inhibit aromatase in a reversible (competitive) manner and are more effective [72]. The general mode of action of aromatase inhibitors has been suggested to be due to the coordination of the inhibitor with the iron atom of the catalytic heme group within aromatase [73]. The aromatase catalytic cleft contains the amino acids Met 374 from the b3 loop; Leu 477 and Ser 478 from the b8–b9 loop; Ile 133 and Phe 134 from the B–C loop; Val 370, Leu 372, and Val 373 from the K-helix-b3 loop; Ile 305, Ala 306, Asp 309, and Thr 310 from the I-helix; and Phe 221 and Trp 224 from the F-helix [33]. 7-dehydrobrefeldin A, 4′-epialtenuene, and atransfusarin, which exhibited the highest affinity to human aromatase cytochrome P450 active site, were screened for their drug-likeness and ADME pharmacokinetics using the website servers [43,74] and showed promising GIT absorption and water solubility.

## 5. Conclusions

The ethyl acetate extract from endophytic *Alternaria tenuissima* AUMC14342 showed anticancer potential against HeLa (cervical cancer), SKOV-3 (ovarian cancer), and MCF-7 (breast adenocarcinoma) cell lines in an SRB assay used to assess cancer cell viability. Cell cycle distribution analysis of treated MCF-7 cells revealed cell cycle arrest at the S phase with a significantly increased cell population. Bioactive secondary metabolites in the EtOAc extract were characterized using LC-MS/MS, and their molecular docking analysis against human placental aromatase exhibited a promising affinity to the aromatase active site. Moreover, ADME pharmacokinetics and drug-likeness properties of 7-dehydrobrefeldin A, 4′-epialtenuene, and atransfusarin revealed good GIT absorption and water solubility. The current research will enrich and enhance the bioprospecting of anticancer natural metabolites from endophytic fungi to achieve a sustainable supplement for safe chemotherapy.

**Supplementary Materials:** The following supporting information can be downloaded at: https://www.mdpi.com/article/10.3390/cimb44100344/s1, mass chromatograms of identified compounds (Figures S1–S21).

**Author Contributions:** Conceptualization, M.E.A. and A.M.A.H.; methodology, M.E.A., A.A.A.M., and A.M.A.H.; software, M.E.A. and F.S.A.-K.; validation, A.A.A.M., M.E.A., and N.F.A.-D.; formal analysis, N.D.D.; investigation, M.E.A. and H.S.A.; resources, M.E.A. and A.M.A.H.; data curation, A.A.H.; writing—original draft preparation, F.S.A.-K., A.A.H., H.S.A., and N.D.D.; writing—review and editing, M.E.A., A.A.A.M., and A.M.A.H.; visualization, A.M.A.H. and M.E.A.; supervision, A.A.A.M. and N.F.A.-D.; project administration, A.A.A.M. and N.F.A.-D.; funding acquisition, A.A.A.M. All authors have read and agreed to the published version of the manuscript. **Funding:** This research was funded by the Deanship of Scientific Research at King Saud University through research group No. RG-1441-419 and the Deanship of Scientific Research at King Saud University logistic support through the Research Assistant Internship Program, Project No. RAIP-4-21-2.

**Institutional Review Board Statement:** Not applicable.

**Informed Consent Statement:** Not applicable.

**Data Availability Statement:** Not applicable.

**Acknowledgments:** The authors extend their appreciation to the Deanship of Scientific Research at King Saud University for funding this work through research group No. RG-1441-419. The authors also extend their appreciation to the Deanship of Scientific Research at King Saud University for the logistic support of this work through the Research Assistant Internship Program, Project No. RAIP-4-21-2.

**Conflicts of Interest:** The authors declare no conflict of interest.

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
