# Peer review of "Cytotoxic Potential of Alternaria tenuissima AUMC14342 Mycoendophyte Extract: A Study Combined with LC-MS/MS Metabolic Profiling and Molecular Docking Simulation"

_cimb, doi:10.3390/cimb44100344_

Round 1

Reviewer 1 Report

Dear authors,

The study of the Alternaria tenuissima AUMC14342 Mycoendophyte extract using experimental and computational investigations is very interesting and pleasing to read. The subject is also very relevant.

Here are my comments and a few suggestions on what can be improved to enrich the paper:

*GENERAL*

- Please check the Alternaria tenuissima abbreviation. Many times, the text is not in italics or correctly abbreviated.

- Please increase the resolution of most of the figures. Some are difficult to see or look a bit blurry.

- Please check Tables and Figures crossing between pages. Also, on Tables (like Table 2 or Table 3), add space before and after the ± symbol.

*INTRODUCTION*

- Page 2: “Computational chemistry methods have been successfully applied to investigate the 84 chemical reactions and binding patterns of a wide variety of biological and chemical 85 systems [27]” – only one paper is cited as a reference. I suggest the authors mention other studies, particularly more recent ones, and maybe comment on the advantages/disadvantages of the methodology. Some suggestions: https://doi.org/10.1016/j.jtcme.2021.05.001,  https://doi.org/10.3390/membranes10060130, https://doi.org/10.2174/1568026620666201204155830, https://www.nature.com/articles/s41598-021-83626-x

 *METHODS*

- Page 3:

Please add the reference for PDB structure; Triangle Matcher, London dG and GBVI/WSA dG.

Also, please include the detailed parameters chosen to determine the docking protocol.

Was the protocol validated? If yes, please add the reference. If not, please add an explanation of how it was validated.

Please add references for Molinspiration and SwissADME servers. Which parameters were selected?

*RESULTS AND DISCUSSION*

- Figure 1: please increase the resolution of the figure and increase the axis units. Too small. Also, check the format because it is crossing pages.

- Table 3: which unit? %?

- Figure 6 is difficult to see. Suggest making it larger and with higher contrast.

- Table 5: how the energy of the conformer is calculated? And how feasible is that result compared to experimental measurements? Does it also relate to Pose Score or RMSD?

- Pages 11-12: How do the H-bonds relate to the results observed? Because it looks like they are not determinant or direct related to the posing score.

- Also, how to confirm those posing scores are good enough to determine the best interaction? Would molecular dynamics not be better for verifying the stability of each compound?

- Table 6: units?

Author Response

Response to Reviewer 1 Comments

The study of the Alternaria tenuissima AUMC14342 Mycoendophyte extract using experimental and computational investigations is very interesting and pleasing to read. The subject is also very relevant.

Here are my comments and a few suggestions on what can be improved to enrich the paper:

*GENERAL*

- Please check the Alternaria tenuissima abbreviation. Many times, the text is not in italics or correctly abbreviated.

Response: All abbreviations were checked and corrected.

- Please increase the resolution of most of the figures. Some are difficult to see or look a bit blurry.

Response: The resolution of the figures was properly revised.

- Please check Tables and Figures crossing between pages. Also, on Tables (like Table 2 or Table 3), add space before and after the ± symbol.

Response: Checked and corrected.

*INTRODUCTION*

- Page 2: “Computational chemistry methods have been successfully applied to investigate the 84 chemical reactions and binding patterns of a wide variety of biological and chemical 85 systems [27]” – only one paper is cited as a reference. I suggest the authors mention other studies, particularly more recent ones, and maybe comment on the advantages/disadvantages of the methodology. Some suggestions: https://doi.org/10.1016/j.jtcme.2021.05.001,  https://doi.org/10.3390/membranes10060130, https://doi.org/10.2174/1568026620666201204155830, https://www.nature.com/articles/s41598-021-83626-x

Response: Thanks for your valuable comments and suggestions; they were considered and stated properly in the revised manuscript.

 *METHODS*

- Page 3:

Please add the reference for PDB structure; Triangle Matcher, London dG and GBVI/WSA dG.

Response: A proper reference was added in the revised manuscript.

Also, please include the detailed parameters chosen to determine the docking protocol.

Response: A Detailed procedure was included in the experimental section.

Was the protocol validated? If yes, please add the reference. If not, please add an explanation of how it was validated.

Response: The protocol was validated after protein preparation by running redocking of the complexed inhibitor to the active site and the RSMD value were 0.18 Å. Complexed ligand and redocked one overlay were shown in supplementary material (Supplementary figure S1).

Please add references for Molinspiration and SwissADME servers. Which parameters were selected?

Response: Suitable references were added. Molinspiration web server were used to determine the molecular properties and SwissADME websites server was employed to calculate the drug likeliness, ADME, and pharmacokinetic parameters of the identified metabolites.

*RESULTS AND DISCUSSION*

- Figure 1: please increase the resolution of the figure and increase the axis units. Too small. Also, check the format because it is crossing pages.

Response: The resolution was improved and the format was adjusted.

- Table 3: which unit? %?

Response: Sorry for this mistake; it is % and was added in the revised manuscript.

- Figure 6 is difficult to see. Suggest making it larger and with higher contrast.

Response: Thanks for your suggestion; it was properly revised.

- Table 5: how the energy of the conformer is calculated? And how feasible is that result compared to experimental measurements? Does it also relate to Pose Score or RMSD?

Response: The MOE dock tools generate the energy of the conformer calculation through the docking experiment and both pose score and RMSD are consider in hit ligand selection. Further the lower posing score and RSMD, the more probability of ligand to interaction with protein experimentally.

- Pages 11-12: How do the H-bonds relate to the results observed? Because it looks like they are not determinant or direct related to the posing score.

Response: The determination of H-bonds was obtained by analysis through reports tool in the MOE ligand interaction software and the posing score is depending on many parameters rather than H-bond like the hydrophobic interactions, distance of H-bond and fitting in the active site pocket.

- Also, how to confirm those posing scores are good enough to determine the best interaction? Would molecular dynamics not be better for verifying the stability of each compound?

Response: The posing scores were confirmed According to the RSMD, fitting in the pocket and interaction with the essential amino acid residue Met374.

Currently running research are working on isolation and purification of selected compounds to investigate the cytotoxicity and inhibition activity against aromatase experimentally for verification of results and development of scaffold for potent compounds.

- Table 6: units?

Response: Thanks for your valuable comment; molecular weight unit was added, the other results are numerical scores value.

Reviewer 2 Report

This study combined both experimental and computational techniques to address the possibility of extracted compounds for inhibiting an aromatase protein, along with their potential for drug development. 

In my opinion the manuscript is suitable for publication after minor questions on molecular docking score is addressed:

Could author describe (either in methodology or at the beginning of results section) the definitions for ‘pose score’ and ‘E. conf’? 

Both quantities were the energy in kcal/mol unit and should describe potential energy of the interactions between the receptor and the ligand. Should E.conf also be a potential energy? Why are they mostly ‘positive’ for receptor-ligand systems in bound state?

Please also explain why authors judged the compounds by E.conf, not pose score. 

Author Response

Response to Reviewer 2 Comments

This study combined both experimental and computational techniques to address the possibility of extracted compounds for inhibiting an aromatase protein, along with their potential for drug development. 

In my opinion the manuscript is suitable for publication after minor questions on molecular docking score is addressed:

- Could author describe (either in methodology or at the beginning of results section) the definitions for ‘pose score’ and ‘E. conf’? 

Response: Thanks for your valuable suggestion; definitions were adressed as possible in the experimental section.

- Both quantities were the energy in kcal/mol unit and should describe potential energy of the interactions between the receptor and the ligand. Should E.conf also be a potential energy? Why are they mostly ‘positive’ for receptor-ligand systems in bound state?

Response: Thanks for your comment;  an important feature of organic compounds is that they are not static, but rather have conformational freedom by rotating, stretching, and bending about bonds. Each different arrangement of the atoms in space is referred to as a "Conformer." Different conformers can have vastly different energies, and the relative proportion of each conformer is related to the energy difference between them. Despite ligand energy minimization, after docking in rigid receptor-flexible ligand protocol the conformation of the ligand compound may be changed and as a result of its rotation or strain to fit in the active site and the lower the energy the more the stability of conformer in the active binding site.

Ref.:  Sein, L.T.;  Varnum, J.M. and Jansen, S.A. Conformational modeling of a new building block of humic acid: approaches to the lowest energy conformer. Environ. Sci. Technol. 1999, 33, 4, 546–552. https://doi.org/10.1021/es9805324

- Please also explain why authors judged the compounds by E.conf, not pose score.

Response: We judged the compounds mainly depending on pose score and RSMD, fitting in the pocket and interaction with the essential amino acid residue Met374.
